



Geoscientific
Model Development

# STORM 1.0: a simple, flexible, and parsimonious stochastic rainfall generator for simulating climate and climate change

**Michael Bliss Singer**[1,3]**, Katerina Michaelides**[2,3]**, and Daniel E. J. Hobley**[1]

[1]School of Earth and Ocean Sciences, Cardiff University, Cardiff, UK
[2]School of Geographical Sciences, University of Bristol, Bristol, UK
[3]Earth Research Institute, University of California Santa Barbara, Santa Barbara, CA, USA

**Correspondence:** Michael Bliss Singer (bliss@eri.ucsb.edu)

**Abstract.** Assessments of water balance changes, watershed response, and landscape evolution to climate change require representation of spatially and temporally varying rainfall fields over a drainage basin, as well as the flexibility to simply modify key driving climate variables (evaporative demand, overall wetness, storminess). An empirical–stochastic approach to the problem of rainstorm simulation enables statistical realism and the creation of multiple ensembles that allow for statistical characterization and/or time series of the driving rainfall over a fine grid for any climate scenario. Here, we provide details on the STOchastic Rainfall Model (STORM), which uses this approach to simulate drainage basin rainfall. STORM simulates individual storms based on Monte Carlo selection from probability density functions (PDFs) of storm area, storm duration, storm intensity at the core, and storm center location. The model accounts for seasonality, orography, and the probability of storm intensity for a given storm duration. STORM also generates time series of potential evapotranspiration (PET), which are required for most physically based applications. We explain how the model works and demonstrate its ability to simulate observed historical rainfall characteristics for a small watershed in southeast Arizona. We explain the data requirements for STORM and its flexibility for simulating rainfall for various classes of climate change. Finally, we discuss several potential applications of STORM.

## 1 Introduction

Models of watershed response (rainfall–runoff), water balance, and landscape evolution require characterization of the driving climate, particularly spatially explicit rainfall fields over a time series. The spatial and temporal variability in water delivery to the land surface from the sky and its fate within a drainage basin are a major control on

a. the partitioning of water between infiltration and runoff, which affects flood risk and water resources (Beven and Freer, 2001; Beven et al., 1995; Michaelides and Wilson, 2007; Slater et al., 2015; Liang et al., 1994);

b. water availability to vegetation, which impacts growth, survival, ecosystem health, and the carbon cycle (Rodriguez-Iturbe et al., 2001; Peñuelas et al., 2011; Singer et al., 2014; Caylor et al., 2006); and

c. patterns and processes of sediment erosion and deposition, which affect the long-term evolution of landscapes (Singer, 2010; Hobley et al., 2017; Tucker and Bras, 2000; Slater and Singer, 2013; Tucker and Slingerland, 1997; Michaelides et al., 2018), as well as the redistribution of contaminants with basins (Singer et al., 2013; Springborn et al., 2011; Higson and Singer, 2015) and the hydrologically controlled in situ biogeochemical processing of such contaminants into more dangerous forms (Singer et al., 2016; Donovan et al., 2016a, b).

However, generating realistic, spatially explicit rainfall fields in drainage basins is a major challenge for several reasons. First, rain gauge data are not typically available at the

**Published by Copernicus Publications on behalf of the European Geosciences Union.**

appropriate spatial representativeness or length of record to well characterize spatial heterogeneity, although radar data can be helpful to improve spatial representation of storms. Second, rain gauge data represent only one realization of multiple potential temporal sequences and spatial patterns of rainfall. Third, global climate models (general circulation models, GCMs) operate at spatial resolutions that are too coarse to represent heterogeneous rainfall fields over small basins or rainfall intermittency (Grotch and MacCracken, 1991; Trenberth et al., 2017). Fourth, while weather generators (regional circulation models, RCMs, or convection permitting models, CPMs) can downscale GCM output for use in dynamic simulation of weather, the model interaction is unidirectional and a regional model is wholly reliant on the boundary conditions provided by the GCM, so it may not well characterize the regional dynamics of climate change (Prein et al., 2015, 2017; Endris et al., 2013; Dunning et al., 2017). Fifth, weather generators that operate at high spatial resolution based on the relevant physics of air mass movement and precipitation formation require detailed information on winds and storm trajectories (Skamarock et al., 2008) that are not available for most basins, and which are challenging to summarize over longer periods of time (e.g., decades).

A further consideration is that internationally agreed climate change scenarios themselves (e.g., the Intergovernmental Panel on Climate Change, IPCC) may limit our learning about the regional expression of recent and future climate change because they have already constrained the problem into covarying sets of gridded climate variables from GCM output (or reanalysis data products) based on particular global emissions scenarios. Given major challenges of GCMs to represent rainfall at the basin scale with variance in topography and orography, ensemble output of climate change projections from GCMs (e.g., CMIP5, CMIP6) is unlikely to provide good characterization of the regional or local expression of climate change (Dunning et al., 2017). More importantly, GCM output and reanalysis data products do not allow for the flexibility to assess watershed responses to a wide range of potential regional climate changes that could impact runoff/flood regimes, groundwater recharge, the water balance between plants and the hydrologic cycle, and basin-wide erosion and topographic development. This is especially true in regions where orography and other complicating land surface dynamics affect rainfall fields.

There is an existing class of rainfall space–time rainfall generator models, which are capable of creating spatially explicit rainfall fields for various purposes (Paschalis et al., 2013; Peleg and Morin, 2014; Niemi Tero et al., 2015; Peleg et al., 2017; Benoit et al., 2018). However, each has its peculiarities in operation, data requirements, computational efficiency, programming language, and output resolution, limiting general applications of rainfall simulation to a wider range of modeling applications such as watershed response, ecohydrology, geomorphic landscape evolution, and land-surface responses to a changing climate.

To fill this research gap, we have developed the STOchastic Rainfall Model (STORM), which generates time series of spatially explicit rainfall fields over a gridded domain and spatially uniform time series of evaporative demand. Together, these time series can be used to drive watershed responses within dynamic rainfall–runoff models, water balances within land surface models, or runoff/erosion regimes within landscape evolution models. STORM was introduced elsewhere to explore rainfall patterns and processes within a small drainage basin in the southwest US (Singer and Michaelides, 2017), but here we provide the relevant detail about the model initialization, operation, and evaluation, and describe various improvements that have been made since its initial appearance in the scientific literature. Additionally, we suggest several additional modifications to STORM that may improve its long-term utility, and we outline its potential for broader use in various hydrologic, ecohydrologic, and geomorphic applications. We also provide links to open-source code for STORM in two forms (Matlab and Python), along with sample input data and parameters.

## 2 Model initialization and operation

STORM is an empirical–stochastic rainfall generator designed for simple, heuristic simulation of high-resolution drainage basin rainfall under control climate conditions or under different classes of climate change. The term "empirical–stochastic" (sensu, Singer and Dunne, 2004) refers to Monte Carlo selection of several key rainstorm characteristics from distributions that are created from historic datasets. STORM performs this multilayer parameter selection to create multiple sequences of spatially varying rainfall over a drainage basin and over a multidecadal time series. STORM output is particularly useful for simulating rainfall patterns in regions subjected to convective rainfall, where gridded datasets of precipitation do not capture the dynamic behavior (spatial and temporal variability) of rainstorms, and where GCM output is of limited utility in understanding expected changes in rainfall regimes over a basin.

Implementing STORM involves initialization and operation steps (Fig. 1). Initialization includes the CE2 creation of distributions of relevant variables that characterize rainstorms and the rainfall they produce over a spatial grid. STORM can be run with one or more hydrologic seasons with different rainfall characteristics. Model inputs include the following (Fig. 2): CE3 annual/seasonal precipitation total, $P_{\text{Total}}$ (Fig. 2a); storm area, which determines which grid locations in the basin are "hit" by each storm (Fig. 2b); storm center location on a storm grid with defined spacings (Fig. 2c); storm duration, $P_D$ (Fig. 2d); rainfall intensity–duration curve number based on a family of curves from which intensity, $P_I$, at the storm center is determined based on the selected value of $P_D$ (Fig. 2e); and storm spatial gradient, or the decline in $P_I$ with distance from the storm center

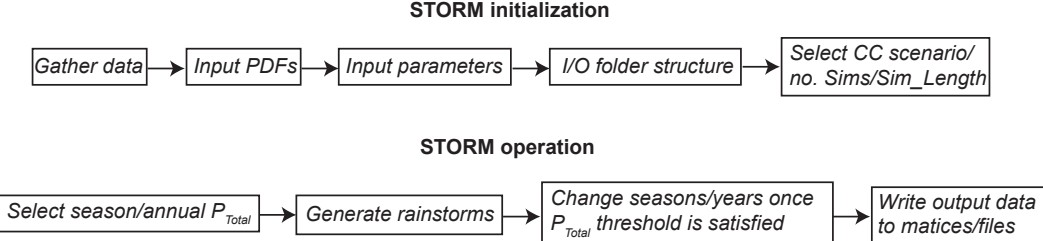

**Figure 1.** Schematic flow diagram illustrating key steps in STORM initialization and operation. Initialization involves gathering data, creating input PDFs, generating input parameters, creating and input/output (I/O) folder structure, and selecting a scenario of climate change, as well as the number of simulations and the length of each simulation. Operation proceeds with the selection of a seasonal or annual precipitation total threshold, followed by generation of rainstorms until the total threshold is satisfied, at which point the season or year changes. Finally, output data are written matrices and files. These steps are outlined in detail within the source code documentation: https://github.com/blissville71/STORM (last access: 6 September 2018).

(Fig. 2f). An additional distribution that is required to create a complete dataset for driving other models is potential evapotranspiration (PET). The PET distribution contains historic daytime and nighttime values for the region of interest organized by month of the year. This collection of values is sampled on a twice-daily basis to characterize average daytime and nighttime evaporative demand that follows the same time signature as the gridded rainfall. PET is assumed to be spatially uniform across the watershed for any 12 h period. Each of these initializing variable distributions can be created from data from a drainage basin, should these data be available (e.g., from one, several, or a network of rain gauges and meteorological stations), or theoretically, based on nearby basin data or generalizing assumptions. Finally, characterizing interarrival times between storms is necessary to enable explicit watershed responses to rainfall inputs (Fig. 3). A distribution of interarrival times is assembled from historic data at all relevant gauging stations. The probability density functions (PDFs) that represent these historical data for each model variable reside external to the STORM code which calls them, and they can thus be easily modified as new information becomes available. For this paper, PDFs were fit manually using Matlab's distribution fitting tool (distfittool), but we recommend that this be automated using a code that optimizes the fit based on maximum likelihood estimators. The particular distributions shown in Fig. 1 were generated by manual investigation based on best fit, but we recommend the automated approach. TS1

Intensity–duration curves were generated (annually or for each season) by first computing the maximum $P_I$ value for each value of $P_D$ (per minute). We then smoothed the resulting points by locally weighted scatterplot smoothing (LOWESS; Cleveland, 1979) using the "smooth" function in Matlab with a span of 0.1 with a second-order polynomial model. Subsequently, we fit a distribution to the LOWESS curve using the curve fitting tool in Matlab. This resulted in the functional form of a double negative exponential curve, consistent with other datasets from dryland regions (Nichol-

son, 2011). The general equation for the $P_I$–$P_D$ curves in Fig 2e is $P_I = \lambda \times \exp(-0.508 \times P_D) + \kappa \times \exp(-0.008 \times P_D) + c$. Parameter values for curve 1 are $\lambda = 642.2$; $\kappa = 93.1$; $c = 4.5$. Next, we maintained the same functional form of the fitted maxima curve (curve 1) but decreased the magnitude of its coefficients (not the decay parameters) and the intercept, where relevant, by percentiles (90th, 80th, 70th, ... 10th, 5th) to generate multiple curves that occupy the full phase space of measured $P_I$–$P_D$ pairs (Fig. 2e). Note: $P_I$ values are selected from one of the curves in Fig 2e using a selected value of $P_D$ from Fig 2d, and then a fuzzy tolerance within $\pm 5$ mm h$^{-1}$ is applied.

The phase space of $P_I$–$P_D$ is not uniform in terms of probability of occurrence of each rainfall event. There is a tendency for more intense storms to occur less frequently than less intense ones. Therefore, each $P_I$–$P_D$ curve is assigned a probability of selection that reflects this fact, wherein the most intense curves (1–3) are assigned lower probability of selection ($-30$ %, $-20$ %, $-10$ %, respectively) and the least intense ones (9–11) are assigned higher probability of selection ($+30$ %, $+20$ %, $+10$ %, respectively) with respect to a hypothetical uniform distribution in curve selection probability (Fig. 4a). In other words, rather than assigning a uniform selection probability of 0.091 to all 11 curves, we decreased the probability of selection of the top three and increased the probability of selection of the bottom three curves. This yields the adjusted curve of probabilities labeled as control in Fig. 4a. The empirical curve-fitting method described here is not very elegant or generalizable to different areas, so in future versions of STORM, we will explore the use of cupolas to represent intensity–duration relationships as marginal probabilities (e.g., Vandenberghe et al., 2011).

STORM also incorporates orography, or the tendency for rainfall to vary with elevation, which is common in many drainage basins around the world. We simulate orography by further modifying the probability of curve number selection from the control curve to account for higher intensity rainfall at higher elevations (Fig. 4b). Based on a hyp-

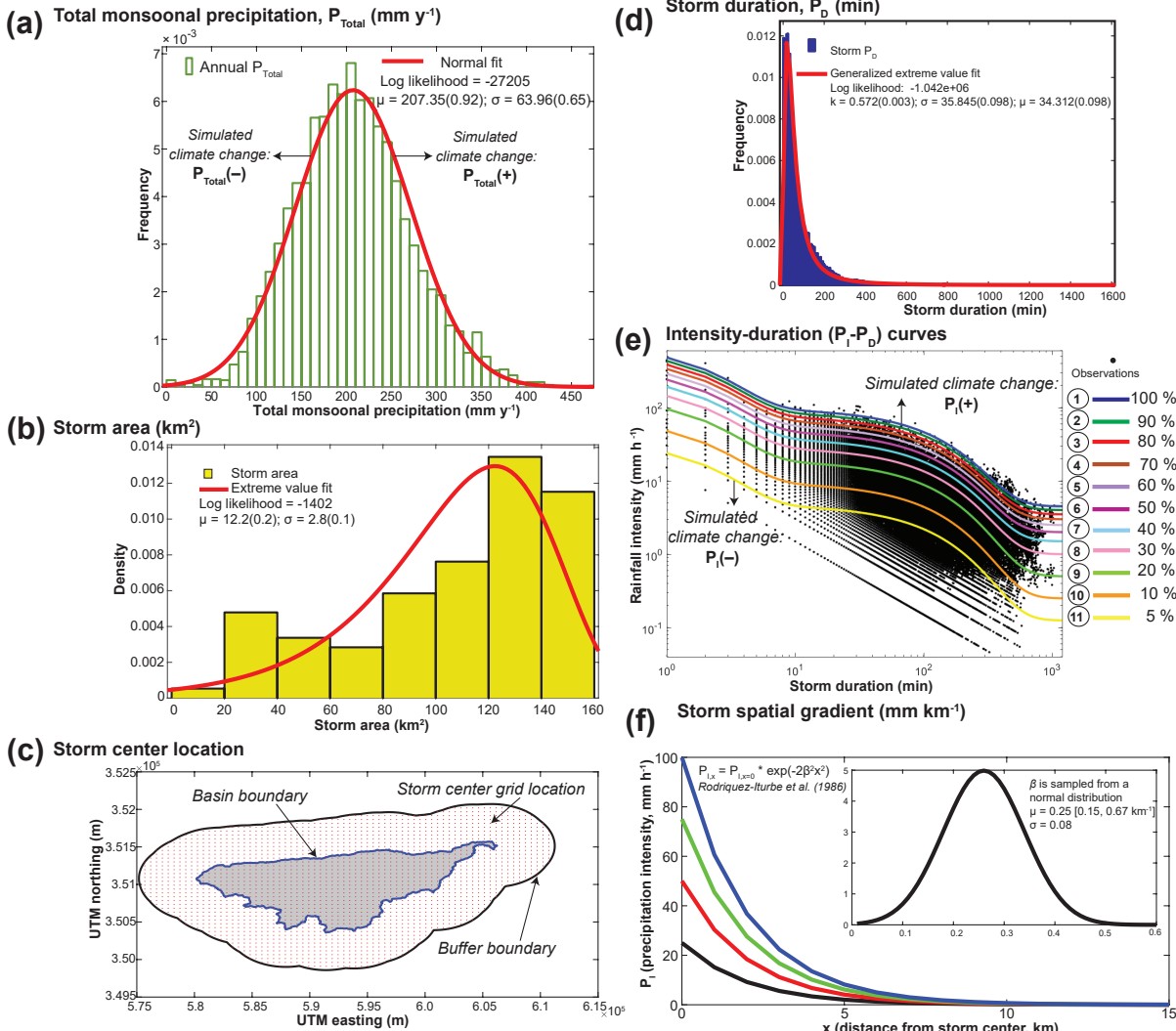

**Figure 2.** Components required for model initialization and simulation. For each year, a selection is made from the PDF of $P_{\text{Total}}$ **(a)**. Then, during each simulation year, selections are made from the following PDFs on a storm-by-storm basis: storm area **(b)**, storm center location on a Universal Transverse Mercator (UTM) grid, red dots, within a 5 km buffer around the watershed boundary **(c)**, storm $P_D$ **(d)**, intensity–duration ($P_I - P_D$) curve number **(e)** based on probabilistic selection favoring less intense storms (see below), and storm intensity gradient with distance from storm center **(f)**.

sometric analysis of rainfall as a function of elevation from gauging records, we divided all rainfall grid locations into three orographic groups based on elevation (Fig. 2c from Singer and Michaelides, 2017). Then we modified $P_I - P_D$ curve selection probabilities for each storm center accordingly. The lowest orographic group (OG1) has the selection probability of curve 1 (most intense curve) decreased by 50 %, while the mid-elevation group (OG2) has a 25 % reduction of probability for curves 1 and 11, and the highest orographic group (OG3) has a 50 % reduction of probability for selecting curve 11 (least intense curve) (Fig. 4b). This simple procedure of increasing/decreasing the probability of storm intensity at the storm center location appears to cap-

ture the general form of orography in the test basin (Fig. 2d from Singer and Michaelides, 2017). However, a more explicit or theoretical method for characterizing the effects of orographic precipitation could replace the current method in STORM (e.g., including wind speed and direction, as relevant). We imagine that the cupola method mentioned above would also be suitable for characterizing orography by fitting separate cupolas to $P_I - P_D$ data from gauges within different bands of elevation in a basin determined by hypsometry.

STORM is implemented as a function in Matlab with the following syntax:

```
STORM(MODE, NUMSIMS, NUMSIMYRS, SEASONS,
PTOT_SCENARIO, STORMINESS_SCENARIO,
```

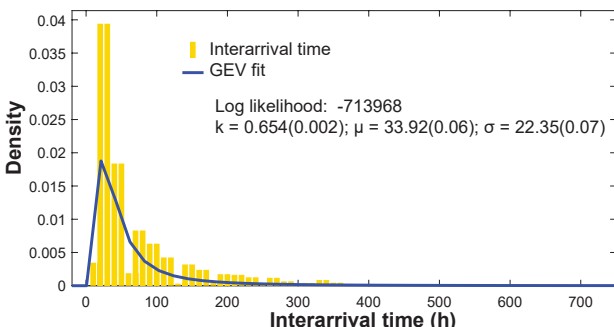

**Figure 3.** PDF of interarrival times which enables watershed response and water balance computations.

```
PTOT_SCENARIO2, STORMINESS_SCENARIO2,
ET_SCENARIO).
```

The input arguments are as follows. MODE refers to whether STORM is running in validation or simulation model. NUMSIMS refers to the number of *n*-year Monte Carlo simulations to be run. NUMSIMYRS is the number of years in each simulation. SEASONS indicates either one or two seasons with different PDFs to sample from. PTOT_SCENARIO refers to the climate change scenario to be simulated with respect to total annual or seasonal rainfall (wetness) in season 1. STORMINESS_SCENARIO refers to the climate change scenario to be simulated with respect to rainfall intensity (storminess) in season 1. PTOT_SCENARIO2 refers to the climate change scenario to be simulated with respect to total annual or seasonal rainfall (wetness) in season 2. STORMINESS_SCENARIO2 refers to the climate change scenario to be simulated with respect to rainfall intensity (storminess) in season 2. ET_SCENARIO refers to climate change scenario for evapotranspiration (evaporative demand). Each of these climate change scenarios for rainfall can be implemented as either step changes (up or down) or as temporal trends (up or down) playing out over multiple decades, and there is full flexibility to modify the magnitude of these changes for both seasons. Currently, the ET climate change scenario only permits step changes. We describe the climate change scenarios in more detail below. Current climate conditions can also be simulated for any or all of these input parameters.

In Python, STORM is implemented similarly but with some syntax differences. It is likewise implemented as a function, which can be simply imported from the defining script. This function is defined as

```
storm(mode, numsims, numsimyrs, seasons,
ptot_scenario, storminess_scenario,
ptot_scenario2, storminess_scenario2,
ET_scenario, storminess_scaling_
factor = 0.05, storm_stepchange = 0.25,
storminess_scaling_factor2 = 0.05,
```

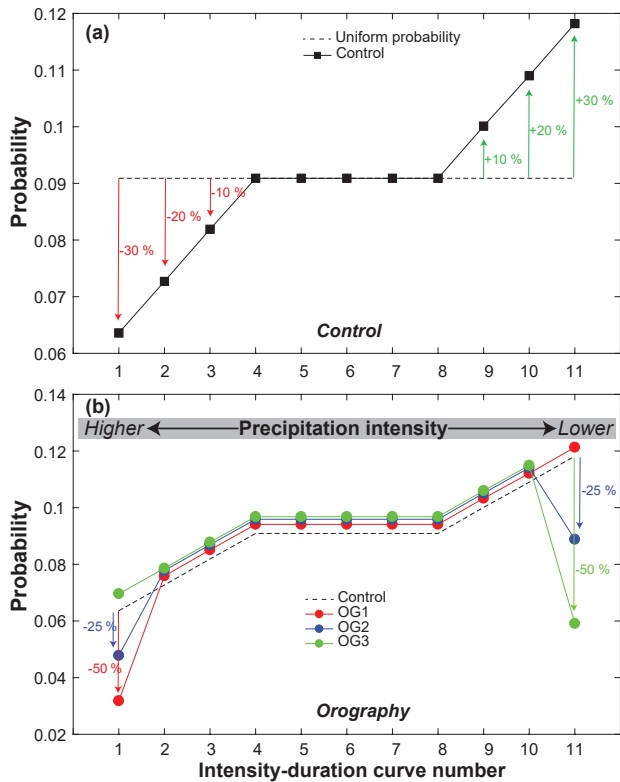

**Figure 4.** Assigned probabilities to $P_I - P_D$ curve numbers for the various simulations. Refer to Fig. 2e for their plotting positions. The initial case is shown in black and represents deviations of $\pm 30\%$, $\pm 20\%$, and $\pm 10\%$ probability for the first/last three curves to reflect the fact that larger storms are less probable than smaller ones **(a)**. This initial case is modified as follows to account for basin orography **(b)**. To represent orographic group 1 (OG1 – lowest elevation group, Fig. 2c from Singer and Michaelides, 2017), we decreased the probability of curve 1 by 50% and applied the difference uniformly to all other curves such that total probability equals 1. To represent orographic group 2 (OG2 – middle elevation group), we decreased the probability of curves 1 and 11 by 25% and applied the difference uniformly to all other curves. To represent orographic group 3 (OG3 – highest elevation group), we decreased the probability of curve 11 by 50% and applied the probability difference uniformly to all other curves. Intensity–duration curve numbers correspond to those listed within circles in Fig. 2e.

```
storm_stepchange2 = 0.25,
ptot_scaling_factor = 0.05,
ptot_scaling_factor2 = 0.05,
ET_scaling_factor = 0.25).
```

The first nine arguments are identical to their Matlab equivalents. The remaining arguments that are supplied with default values permit direct control through the function call of the step and gradual change values specified by the preceding scenarios (see below); in the Matlab version, these are controlled directly from the script defining the function. Documentation is provided as a docstring to the

function. The Python version of STORM has a number of very common Python packages as dependencies – os, time, numpy, datetime, matplotlib, scipy, and six. These are typically installed as standard in Python distributions. STORM in Python also requires the pyshp package, which is readily available through package managers including pip and conda. STORM in Python is agnostic between Python 2 and Python 3. In Python, input files (see below) are supplied as comma separated value (csv) files, rather than .mat files, but the contents of these files are identical to those in Matlab. Where distribution objects are provided in Matlab, these are provided to the Python version as simple text files specifying the key parameters of those distributions. More details of both of these input file formats are provided in the docstrings.

To operate STORM, we simulate rainfall on a storm-by-storm basis with a temporal resolution of 1 min at each rainfall output grid location with a resolution of 1 km. The high spatial and temporal resolution enables rich information to be generated from STORM, allowing for detailed assessment of spatial heterogeneity and temporal variance in rainfall fields. We first select a threshold value of $P_{Total}$ for a simulation year from its distribution and then generate rainstorms until the median running total of $P_{Total}$ across all rainfall output grid locations within the basin equals or exceeds the selected threshold value of $P_{Total}$. Then a new simulation year begins and STORM proceeds until the length of simulation (e.g., several decades) is complete. $P_{Total}$ can be designated as either a seasonal or annual total, depending on whether the watershed of interest is characterized by strong seasonality in rainfall. If STORM is implemented with two seasons, separate PDFs of rainstorm characteristics (Fig. 2a, b, d, and e) should be prepared for each season. PDFs for this paper were fit to historical data from the Walnut Gulch Experimental Watershed (WGEW) using the distribution fitting tool within Matlab v2017b. Sample distributions are provided at https://github.com/blissville71/STORM (last access: 6 September 2018).

To model climate change as a step change in wetness (shift in total annual precipitation), we shifted the $P_{Total}$ distribution up $P_{Total}(+)$ and down $P_{Total}(-)$ by 1 standard deviation without changing its shape (Fig 2a). To model climate change as a step change in storminess (shift in intensity for a particular rainstorm duration), we modified the selected intensity for all storms as $P_I \pm \Psi \times P_I$, where $\Psi$ is a fractional change in storm intensity (Fig 2e). Thus, the selected intensity for all rainstorms is raised for $P_I(+)$ and lowered for $P_I(-)$. We have also built into STORM the capability to assess trends in both $P_{Total}$ and $P_I$ by multiplying the selected value of either rainfall variable for each year of simulation by an annual change scalar (see below for details).

The results generated at each rainfall output grid location over all simulations can be statistically analyzed. STORM generates a rainstorm matrix that includes storm no., storm area ($km^2$), storm duration (min), intensity–duration curve no., storm intensity at storm center ($mm\,h^{-1}$), no. of gaug-

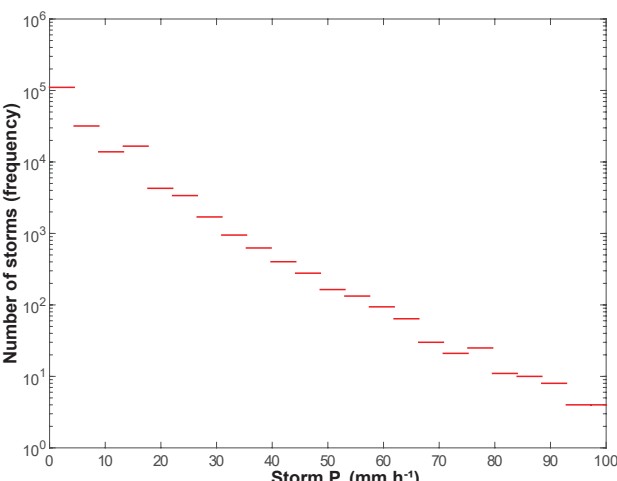

**Figure 5.** Data-derived number of rainstorms for various bins of storm intensity recorded at 85 WGEW gauges over 62 years. The historical record shows that are $\sim 14\,000$ events of $10\,mm\,h^{-1}$, $\sim 3500$ events of $25\,mm\,h^{-1}$, $\sim 200$ events of $50\,mm\,h^{-1}$, and $\sim 4$ events of $100\,mm\,h^{-1}$.

ing (or rainfall grid) locations hit, intensity recession value ($mm\,h^{-1}\,km^{-1}$), storm total (mm), longitude (m), latitude (m), year (yr), and cumulative simulation time (h). STORM also generates a separate matrix of output at each rainfall grid location that includes year, storm no., local storm intensity ($mm\,h^{-1}$), storm duration (min), local storm total (mm), annual local cumulative precipitation total (mm), interarrival time between storms (h), and cumulative simulation time (h).

All simulations of STORM for this paper were done using Matlab v.2017b on an Intel Dual Core i7-4712HQ CPU at 2.30 GHz running on a Dell Inspiron laptop with 16 GB of RAM. In simulation mode, each simulation of 30 years on a grid of 128 rainfall locations within a 149 $km^2$ basin took $\sim 12.5$ min for the control run, or $\sim 25$ s per simulated year. The computational time of each ensemble varies depending on the scenario. For example, the simulation of $P_{Total}(+)P_I(-)$ took $\sim 20$ min per 30-year simulation (or $\sim 40$ s per simulated year).

## 3 STORM application to Walnut Gulch

Here, we provide a few more details about how we assembled the relevant PDFs for WGEW that are shown in Fig. 2. PDFs were generated within Matlab using the distribution fitting tool, though the methods are straightforward enough to be easily applied within many data analysis software packages. To create the PDF of storm areas, we truncated the distribution of Syed et al. (2003) to exclude all rainstorms in their largest spatial bin (which comprised approximately one-half of all their rainstorms) and then fitted an extreme value distribution to the remaining storms (Fig. 2b). Syed et al. (2003)

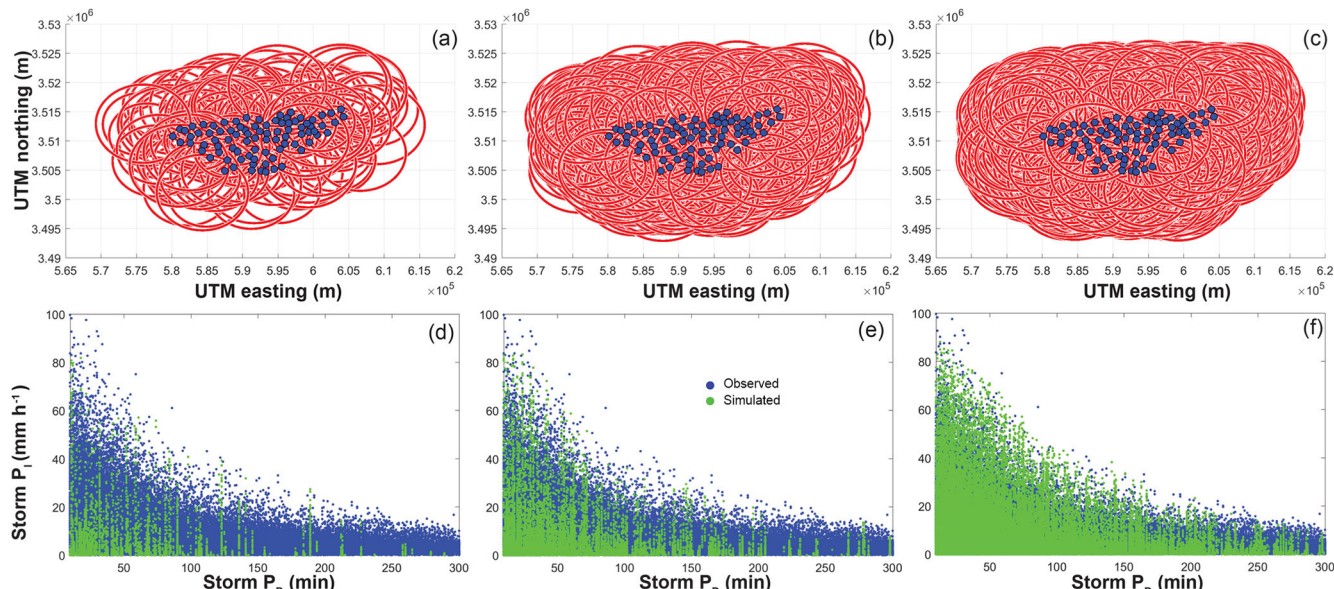

**Figure 6.** Maps of simulated storms (red circles) on the model grid for individual simulations over periods of 1 year **(a)**, 5 years **(b)**, and 25 years **(c)**. Solid blue circles represent WGEW gauging stations. Lower panels indicate the corresponding intensity and duration values for simulated storms within the 1-year **(d)**, 5-year **(e)**, and 25-year **(f)** model runs for part of the intensity–duration phase space. Green dots represent simulated storms and blue dots represent gauge observations (63 years of per storm data at 85 gauging locations).

indicated that this bin of high frequency for large spatial areas is probably an artifact of sampling such that they are overestimating storm areas due to repeat sampling of moving storms and due to the occurrence multiple simultaneous storm cells. Thus, our exclusion of this largest area bin is justified to achieve our goal of modeling individual (discrete) rainstorms and the resulting distribution has a mean storm area of just under $90\,\mathrm{km^2}$. In our initial tests of STORM, we found that the results were very sensitive to the $\beta$ parameter (Eq. 1; Fig. 2f). Thus, to enable a variety of plausible storm gradients, we allowed $\beta$ to vary according to a normal distribution ($\mu = 0.25$, $\sigma = 0.08$, truncated to the interval [0.15, $0.67\,\mathrm{km^{-1}}$]), based on the range of values reported elsewhere for WGEW (Eagleson et al., 1987; Morin et al., 2005).

We analyzed all the rainfall data from WGEW to assess the frequency of different storm intensities and found that, as expected, there is a high frequency of rainstorms with low intensity, and vice versa (Fig. 5). This validates our development of the control set of curve selection probabilities (Fig. 4a).

In each multiyear simulation, storms of various areas cover different parts of STORM's model domain. If we aggregate all the simulated rainfall data from each of the rainfall output grid locations, we see increasing coverage of the $P_I - P_D$ phase space with increasing length of simulation in years (Fig. 6). Simulations of 25 years appear to fill in much of the $P_I - P_D$ phase space for control (no-climate-change) conditions (Fig. 6f). This indicates that STORM is faithfully representing the input data on intensity–duration of rainfall in a stochastic treatment over multiple decades.

Finally, a plausible characterization of climate change is necessary to gain insight into the potential impacts to watershed response. In terms of precipitation, climate change can manifest in wetter or drier conditions (e.g., over a season or a year) and/or in different storm characteristics (e.g., relationship between $P_I$ and $P_D$). We therefore suggest there are two classes of climate change that affect convective precipitation and watershed response. In STORM, we can simulate different classes of climate change based on very simple rules. To simulate step changes in basin wetness (annual/seasonal precipitation totals), we shift the PDF of $P_{\mathrm{Total}}$ up for $P_{\mathrm{Total}}(+)$ or down for $P_{\mathrm{Total}}(-)$ by 1 standard deviation, while retaining the same shape of the distribution (Fig. 2a). Note: this truncates the left tail of original $P_{\mathrm{Total}}$ distribution for the scenario of $P_{\mathrm{Total}}(-)$. To simulate step changes in storminess, we multiply the selected value of $P_I$ at the storm center by a scalar fraction and add (subtract) the product to the selected $P_I$ value to reflect increased (decreased) storminess (Fig. 2e). Specifically, we modified the selected intensity for all storms as $P_I \pm \Psi \times P_I$, where $\Psi$ is a fractional step change in storm intensity. Thus, the selected intensity for all rainstorms is raised for $P_I(+)$ and lowered for $P_I(-)$. In this paper, we used a step change of $\Psi = 0.25$.

We also enable the simulation of temporal trends in rainfall. We include scalar multipliers that can be set within the STORM code, and which are used to modify the relevant value of rainfall each year. For example, to characterize a

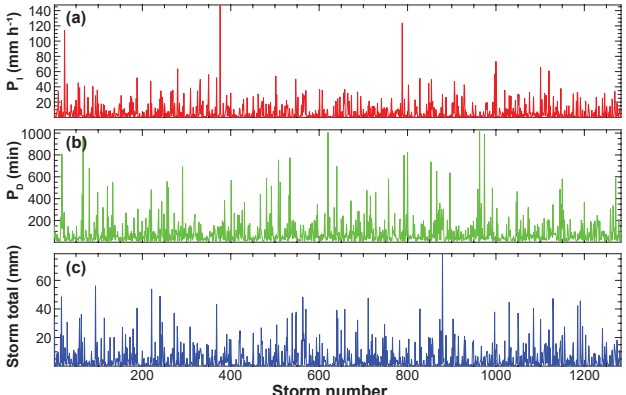

**Figure 7.** Illustration of the detail of model output that is generated on a storm-by-storm basis for one 30-year simulation at a single gauging (rainfall grid) location, showing rainfall intensity **(a)**, duration **(b)**, and storm total **(c)** for each storm event simulated at that gauge.

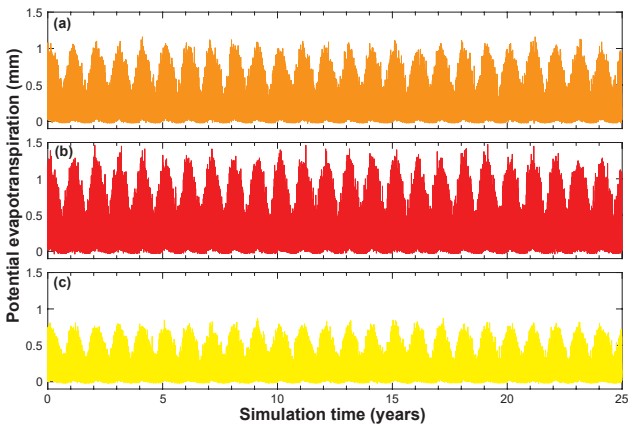

**Figure 8.** Illustration of simulated evapotranspiration for a 25-year simulation with two seasons under **(a)** the control climate change scenario (no change in PET), **(b)** a 25 % step-change increase in PET, and **(c)** a 25 % step-change decrease in PET.

trend in wetness (total precipitation per season or year), we annually update the $P_{\text{Total}}$ PDF by generating a new PDF for each year after progressively modifying the mean value of the PDF (Fig. 2a) as $\mu = \mu \pm \mu \times \Phi$, where $\Phi$ is a fractional scaling factor that increases the PDF of $P_{\text{Total}}$ each year of simulation. Within STORM, a similar procedure is used for trends in storminess. In this case, to characterize a positive trend in storminess (magnitude of storm intensity per season or year) TS2, we annually update the $P_{\text{Total}}$ PDF by generating a new PDF for each year after progressively modifying the mean value of the PDF TS3 as $P_I \pm \Omega \times P_I$, where $\Omega$ is a fractional scaling trend that increases the selected value of storm intensity by an accumulating trend each year of simulation (i.e., the storminess trend in any year of simulation is computed as $\Omega_Y = \Omega \times Y$, where $Y$ is the simulation year). In this paper, we used initial values for both $\Omega$ and $\Phi$ of 0.05. We note that it is possible to simulate trends in storminess separate from trends in wetness, or in combination.

## 4 Model output and evaluation

It is possible to extract from STORM detailed output of rainfall characteristics for discrete rainstorms at each rainfall grid location, including storm-by-storm intensity, duration, and storm totals (Fig. 7). Thus, one can develop a localized time series of these rainstorm characteristics for different locations or sub-basins within a drainage basin. Once the simulated interarrival times between each pair of rainstorms (Fig. 3) are added back to the time series and used alongside the simulated time series of PET (Fig. 8), we obtain a temporally explicit climate driver that can be used to drive models of watershed response, land surface models, or to make localized water balance calculations. Figure 9 illustrates output at a single gauging location of explicit time series of storm

rainfall for different climatic scenarios (see below). Note that the time series shown in Figs. 7–9 only illustrate output for a single multidecadal simulation. The stochastic capability of STORM allows for generation of multiple $n$-year simulations. Thus, the STORM output for all such simulations should ideally be analyzed statistically at each representative location to produce representative time series of rainfall variables. Otherwise, they can be used as ensemble inputs to other model frameworks.

We evaluated the model's skill at simulating observed rainfall characteristics at 85 gauges in WGEW. We selected three representative variables: number of storms per year, average storm total, and total annual precipitation. We compared observed versus simulated values of these variables at each gauging location in WGEW. Figure 10 shows these as simulated versus observed on 1 : 1 plots, and relevant statistics on model skill are provided. Generally, STORM demonstrates a high level of skill at simulating rainfall characteristics across the domain, without any model tuning. The spread of points around the 1 : 1 line arises from the ensemble of multidecadal simulations, and it will therefore vary from ensemble to ensemble. Generally, it is desirable to have a range of values that span the 1 : 1 line (above and below it). This ensures the model is generating a field of rainstorm characteristics beyond those which exist in the historical record, a notable strength of the Monte Carlo procedure implemented in STORM.

Note: the random selection of model parameters (Figs. 2 and 3) was done using the random function in Matlab (v.2016b), when sampling from a PDF (e.g., $P_{\text{Total}}$), and using the datasample function when sampling from a vector of data points (storm center location). The random number generator seed was shuffled prior at the beginning of each Matlab session.

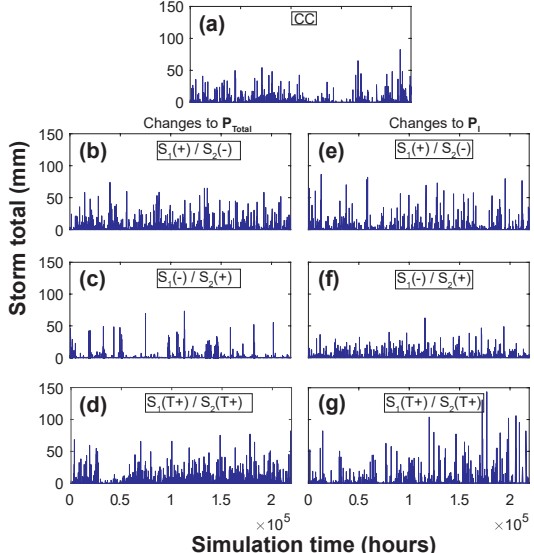

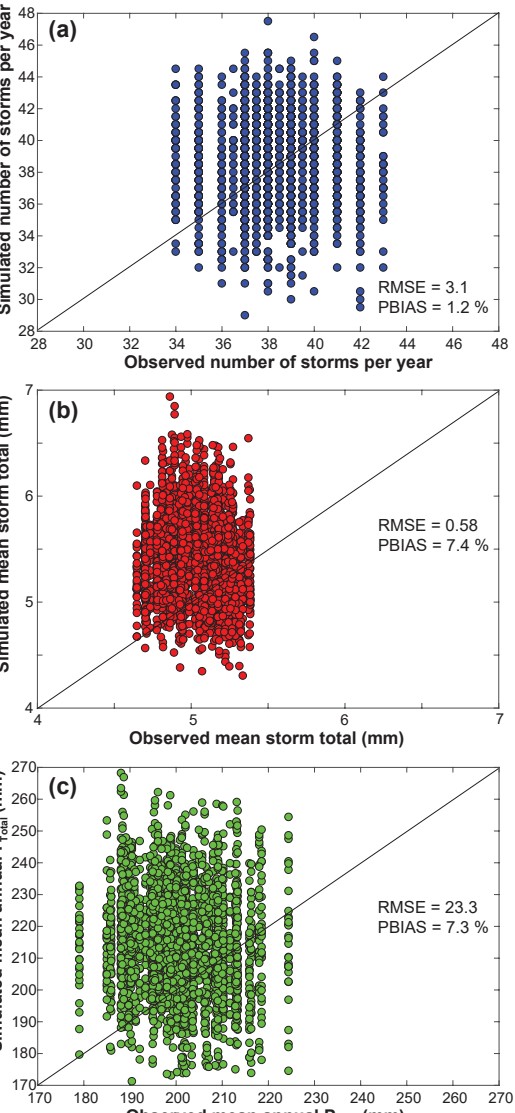

**Figure 9.** Illustration of simulated rainfall at one gauging location for a 25-year simulation with two seasons. The panels are **(a)** control climate conditions (CC); **(b)** a step-change increase in $P_{Total}$ for season 1 and a step-change decrease in $P_{Total}$ for season 2 $(S_1(+) / S_2(-))$; **(c)** a step-change decrease in $P_{Total}$ for season 1 and a step-change increase in $P_{Total}$ for season 2 $(S_1(-) / S_2(+))$; **(d)** positive trends in $P_{Total}$ for both seasons $(S_1(T+) / S_2(T+))$; **(e)** a step-change increase in $P_I$ for season 1 and a step-change decrease in $P_I$ for season 2; **(f)** a step-change decrease in $P_I$ for season 1 and a step-change increase in $P_I$ for season 2; **(g)** positive trends in $P_I$ for both seasons.

## 5 STORM in the context of climate change

Regional gridded datasets provide a picture of trends in certain climate variables that are relevant to rainstorms. Figure 11 shows monthly output at WGEW for the 0.5° CRU TS3.24.01 dataset for temperature anomalies (Fig. 11a) and precipitation anomalies (Fig. 11b). These datasets show a significant increase in temperature of ∼ 2 °C over recent decades (which constitutes the period of the WGEW rainfall record), but there is no clear trend in the monthly precipitation from this dataset. The lack of a trend in monthly precipitation data contrasts with the high-resolution data from WGEW, which showed a long-term increase in $P_{Total}$ but with declining $P_I$ (Singer and Michaelides, 2017). Such changes to the hydrology of a basin could have major implications for its runoff regime, water balance, and landscape evolution. This suggests that gridded global datasets are not adequate for investigations of convective rainfall in dryland (and potentially many other) basins. This observation lends support for both detailed data analysis from dryland datasets such as WGEW but also for modeling approaches such as STORM for exploring the potential impacts of climate change on rainfall and watershed responses.

**Figure 10.** Observed versus modeled statistics on annual monsoon precipitation: median number of storms per year **(a)**, median storm total **(b)**, and mean annual $P_{Total}$ **(c)**. Values of RMSE and PBIAS are also shown. No model tuning was performed to achieve these results. We note that observed values of storm total and $P_{Total}$ are slightly overpredicted by our simulations (over these 30 test ensembles, each of 30 years). Nevertheless, these results demonstrate a high level of model skill in reproducing a range of rainfall characteristics that span observed values without any model tuning.

Figure 8 shows STORM output of ET showing clear day–night variations, as well as seasonality (monthly day–night ET were sampled from historical data compiled by month). Figure 8a shows the control climate scenario, Fig. 8b shows a step-change increase in ET, and Fig. 8c shows a step-change decrease. Figure 9 illustrates STORM rainfall output at a single gauging point in the basin as complete time series that include interarrival times, so the data can be used to drive

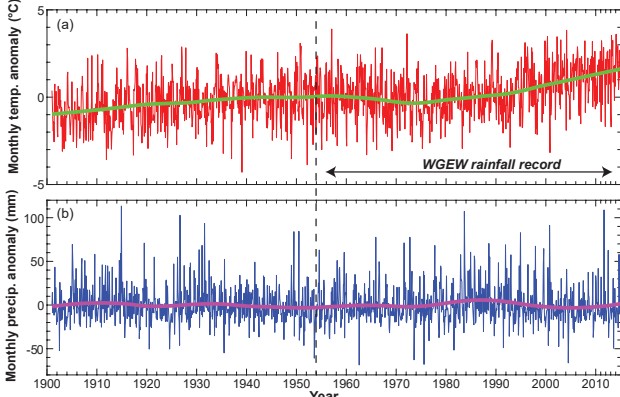

**Figure 11.** Monthly mean temperature **(a)** and monthly mean precipitation anomalies **(b)** for the closest grid location to Walnut Gulch from the 0.5° Climate Research Unit (CRU) TS3.24.01 dataset (Harris and Jones, 2017). These plots show a recent increase in mean temperature that could influence rainfall $P_I - P_D$ relationships, irrespective of the lack of long-term trend in precipitation totals from the gridded dataset. Curves are based on LOWESS smoothing (Cleveland, 1979) via the smooth function in Matlab (v.2016b) using a 30 % data span.

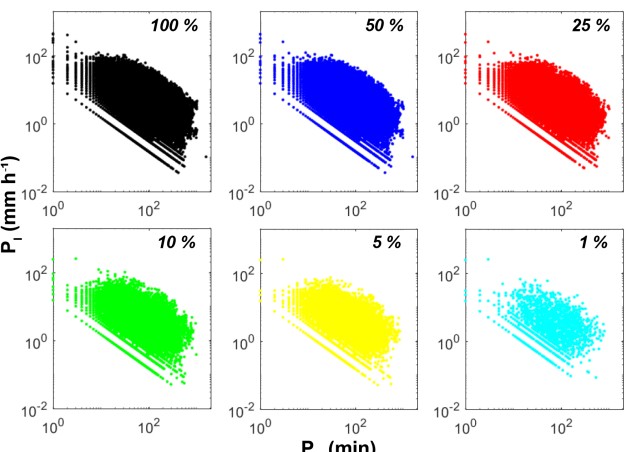

**Figure 12.** Illustration of the limited sensitivity of STORM to the quantity of input data. Plots of $P_I$ versus $P_D$ for various percentages of the complete dataset (indicated at the upper right of each subplot) show that even when only 5 % of the original gauge data are included, the $P_I - P_D$ phase space for the complete dataset is still broadly defined. The number of data points declines across these subpanels from 185 109 (100 %) to 1851 (1 %). These factors indicate that STORM could be reasonably applied in a basin with dramatically less available gauging data, if other storm characteristics can be constrained by other means.

other models. The figure also shows these time series of rainstorm totals for various climate change scenarios including control climate (Fig. 9a), a step-change increase in total season 1 rainfall with a step-change decrease in total season 2 rainfall (Fig. 9b), a step-change decrease in total season 1 rainfall with a step-change increase in total season 2 rainfall (Fig. 9c), positive trends in total rainfall for both seasons (Fig. 9d), a step-change increase in season 1 rainfall intensity with a step-change decrease in season 2 rainfall intensity (Fig. 9e), a step-change decrease in season 1 rainfall intensity with a step-change increase in season 2 rainfall intensity (Fig. 9f), and positive trends in rainfall intensity for both seasons (Fig. 9g). These outputs illustrate the range of capability included in STORM for simulating an array of different regional expressions of climate change that could have important implications for watershed response. In general, increases in $P_{Total}$ tend to densify the number of rainstorms in the time series (and vice versa), while increases in storminess tend to create more peaked rainstorm totals (cf. Fig. 9d, and g).

## 6   STORM data requirements

An important consideration of any model is the data requirements. Obviously, the lower amount of data required enables more widespread model use to tackle environmental problems. However, on the other end of this spectrum, insufficient data can lead to poor model skill. Thus, we aimed to strike a balance that would enable widespread use of STORM with limited data. In terms of storm event rainfall data, we inves-

tigated how much rain gauge data are required by STORM to well characterize the historical phase space of rainfall characteristics. For WGEW, we plotted event rainfall intensity versus duration for 100 % of the available data, 50 %, 25 %, 10 %, 5 %, and 1 % (Fig. 12). This analysis shows that even at 1 % of the available data ($n = 1851$ rainfall events), the $P_I - P_D$ phase space is still clearly delineated, allowing for development of the necessary $P_I - P_D$ curves (Fig. 2e). Therefore, one could use data from a basin with only one or several rain gauges that have collected event rainfall for a few decades.

Another important data requirement in STORM is a PDF of storm areas (Fig. 2b). This may be a more challenging PDF to develop due to limited data. Options here include analyzing spatial statistics on discrete storms from a network of rain gauges (e.g., Syed et al., 2003), analyzing storm characteristics from rainfall radar images (e.g., Peleg and Morin, 2012), or developing a hypothetical distribution based on regional understanding of mapped rainstorms (e.g., from hyetographs). Finally, the density of the gauging network could have important influence on the storm intensity gradient with distance from the storm center, so the parameters of this relationship may be less certain for less dense gauge spacings.

## 7 Extension/modification of STORM for additional applications

For many applications, a complete account of the water balance between precipitation, evapotranspiration, and infiltration is required. Specifically, closing the local water balance requires quantifying the evaporative demand and the length of inter-storm periods, which enable drainage and drying of soil layers. These factors affect the watershed response to subsequent rainfall events. In order to characterize these aspects, we have modified STORM from its original capability (Singer and Michaelides, 2017) to include several new features. First, we have added a PDF of inter-storm periods that is sampled randomly after each storm event. The addition of these inter-storm periods changes STORM output into time series that reflect real time at the Earth's surface (e.g., Fig. 9). Second, we have assembled a PDF of potential evapotranspiration based on measurements of temperature and relative humidity, metrics which are readily available over multiple scales (Fig. 8). Third, we implemented seasonality in rainfall to enable simulations over a single season or year, or over two seasons with distinct differences in precipitation characteristics (distinct PDFs of rainfall in summer compared to winter). The WGEW example implemented for illustration of the model here is one such basin with a strong monsoon season that produces a high percentage of the annual rain and most of the runoff, compared with the winter season dominated by weak frontal storms.

These improvements to STORM now make it suitable as a climate driver of other watershed response models that simulate hydrology between slopes and channels (surface runoff, infiltration, streamflow) (Michaelides and Wainwright, 2002, 2008; Michaelides and Wilson, 2007), groundwater recharge during and after rainfall events (Beven and Freer, 2001), and interactions between streamflow and alluvial aquifers (Evans et al., 2018). It also enables STORM to be useful in water balance models (e.g., land surface models) to assess water availability to plants through dynamic ecohydrological simulation of plant–climate interactions and water utilization (D'Odorico et al., 2007; Caylor et al., 2006; Laio et al., 2006), as well as energy/carbon fluxes between the land surface and the atmosphere (Best et al., 2011; Bonan, 1996). Finally, STORM can also be used to drive geomorphic models that characterize erosion and deposition processes within drainage basins in response to sequences of rainfall and runoff (Michaelides et al., 2009, 2012; Michaelides and Martin, 2012; Michaelides and Singer, 2014), and even landscape evolution models that simulate landform development over longer timescales (Tucker and Hancock, 2010; Hobley et al., 2017). Coupling STORM to such models would enable a wide range of interdisciplinary scientists to investigate key problems in the environment that have their origin in the climate system. These range from which water sources are used by plants (Sargeant and Singer, 2016; Evaristo et al., 2015; Evaristo and McDonnell, 2017; Singer

et al., 2014; Dawson and Ehleringer, 1991) to what is the dominant source and timing of groundwater recharge (Cuthbert et al., 2016; Wheater et al., 2010; Scanlon et al., 2006) to the role of climate in shaping landscape morphology (Singer and Michaelides, 2014; Tucker and Bras, 2000; Tucker and Slingerland, 1997; Michaelides et al., 2018). A version of STORM is under active development in the modular opensource surface process modeling framework Landlab (Hobley et al., 2017), in part to facilitate such future work. Another key area of future work would be to investigate how temporal resolution of rainfall data affects the signal of observed trends in rainfall (e.g., Barbero et al., 2017) and how these might yield different watershed responses.

*Code availability.* Both Matlab and Python versions and sample data can be found at https://github.com/blissville71/STORM (last access: 6 September 2018). Documentation is also provided at that link. The DOI for version 1.0 is https://doi.org/10.5281/zenodo.1291898.

*Author contributions.* MBS and KM developed the idea, MBS wrote and tested the code, MBS wrote the paper with input from KM and DEJH. DEJH translated the code to Python.

*Competing interests.* The authors declare that they have no conflict of interest.

*Disclaimer.* The authors take no responsibility for the use or misuse of the provided code.

*Acknowledgements.* Michael Bliss Singer received funding from the US National Science Foundation Hydrologic Sciences (EAR no. 1700555), Geography and Spatial Sciences Program (BCS no. 1660490), and the US Department of Defense's Strategic Environmental Research and Development Program (no. RC18-C2-1006).

Edited by: Bethanna Jackson
Reviewed by: Nadav Peleg and one anonymous referee

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

## Remarks from the language copy-editor

## Remarks from the typesetter