# Peer review of "STORM 1.0: A simple, flexible, and parsimonious stochastic rainfall generator for simulating climate and climate change"

_Geoscientific Model Development, 2018_

## Short Comment (SC1) · 11 Jun 2018

Dear authors,

in my role as Executive editor of GMD, I would like to bring to your attention our Editorial version 1.1: http://www.geosci-model-dev.net/8/3487/2015/gmd-8-3487-2015.html This highlights some requirements of papers published in GMD, which is also available on the GMD website in the 'Manuscript Types' section: http://www.geoscientific-model-development.net/submission/manuscript_types.html In particular, please note that for your paper, the following requirement has not been met in the Discussions paper:

- "The main paper must give the model name and version number (or other unique identifier) in the title."

Please provide a version number for STORM in the title of your revised manuscript. Note, that a version number is important to identify the status of your developments published in this article.

As explained in our Editorial, articles have to include a code and/or data availability section. The name and the place of the section is predefined, so change the title of your "code and sample data" section accordingly. Referring to the content of the section, the link in the paper does not work.

As explained in https://www.geoscientific-model-development.net/about/manuscript_types.html. GMD is encouraging authors to upload the program code of models (including relevant data sets) as supplement or make the code and data of the exact model version described in the paper accessible through a DOI (digital object identifier). In case your institution does not provide the possibility to make electronic data accessible through a DOI you may consider other providers (eg. zenodo.org of CERN) to create a DOI. Please note that in the code availability section you can still point the reader to how to obtain the newest version. If for some reason the code and/or data cannot be made available in this form (e.g. only via e-mail contact) the "Code Availability" section need to clearly state the reasons for why access is restricted (e.g. licensing reasons).
Especially, please note, that it is not enough, that the code will be available in the future. It must be available now and the exact version of the code published in this article needs to be made available.

As the links to the data in the discussion version of your paper do not work, please upload an author comment stating clearly how the code can be accessed in addition to

a full revision of the Code availability section in the revised version of your article.

Yours, Astrid Kerkweg

---

## Author Comment (AC1) · 18 Jun 2018

Thank you for these comments on our paper. I'm not sure why the pdf version that appeared on GMD did not have active hyperlinks because the Word version I uploaded did. Here is the link to the model and sample data:

https://github.com/blissville71/STORM

We will change the paper title to include the model's version number (1.0) according the editor's suggestion. We will also generate a DOI for this version of the code and provide that. Finally we will revise the section on 'Çode Availability'.

[Figure]

Thank you,

Michael
* * *

---

## Referee Comment (RC1) · N Peleg (Referee) · 22 Jun 2018

In their paper, Singer et al. present the STORM model, a rainfall generator that simulates convective storms for present and future climates. The model is relatively new (earlier version was presented by Singer and Michaelides in 2017), and here the Authors provide additional information regarding the model setting and operation, alongside with several other improvements (e.g. simulating PET). I believe that many hydrologists and geomorphologists will benefit from having a relatively lite (in terms of computational demands and parameterization requirements) rainfall generator model as the one presented here. The paper is well structured and written. Some further information on the model is required, to explain the need in a new rainfall generator and to better understand its engine. I have made several minor suggestions and comments that are listed below.

[page line] or [topic/section]

[Schematic flowchart] A figure illustrating the schematic flowchart of STORM might be useful to understand the model architecture.

[Inputs and parameters] Consider summarizing the required inputs and parameters of STORM in a table.

[Introduction] There are some other space-time rainfall generator models that were recently introduced to the scientific community. For example: STREAP (Paschalis et al., 2013), HiReS-WG (Peleg and Morin, 2014), STEPS (e.g. Niemi et al., 2016), AWE-GEN-2d (Peleg et al., 2017) and a recent stochastic rainfall generator that was presented by Benoit et al. (2018). I suggest adding a paragraph briefly mentioning these models (and others that are similar to STORM, if exists) and explaining why STORM is needed and what functions it can fill, what advantages it has in comparison to the other models, etc.

[2 12-15] I must say I disagree with this statement - using reanlysis data, for example, one can get today a good representation of wind and storm trajectories at fine spatial and temporal resolution, e.g. using MERRA product at 50-km and hourly.

[3 17-18] Why there is a limitation for two seasons? Can the model be used with monthly statistics? What about advection? Is it consider or is the storm stationary in space? Some further information about the cross-correlation between model inputs are required. For example, is the storm area correlated with storm duration? The dependencies between variables need to be discussed.

[3 19] Can a storm have multiple "centers" at a given time step?

[3 25] Does it mean that PET has a different set of statistics also for wet and dry

periods? If the answer is no, then I guess that PET is simulated as a standalone module, i.e. with no correlation to the storm. Is that right?

[page 4] I suggest moving this part to the Supplementary Information or as an Appendix. [4 33] I guess the user can modify the temporal and spatial resolution and the ones given here are the resolution used for the case study. Am I right? If so, I suggest it will be explicitly written. Moreover, here you mention the spatial resolution to be 1-km, but in Figure 1 the example is for 500-m.

[5 8] I cannot access the link. Consider adding a table summarizing the distributions that are fitted for each of the variables (can be as supplementary material). Some of this information I see in Figure 1, but it will be clearer as a separate table.

[5 19] v.2017b

[Figure 1] From the "note" onward: I suggest moving this text from the figure caption to the main text.

[8 6+] Some of the text here explains how the model works and is more suitable to be placed in the methodology section that comes before.

[Model evaluation] What about rainfall extremes? For example, does the model reproduce extreme rainfall intensities satisfactory when comparing model simulation to gauges? Moreover - the model simulates rainfall at the minute scale - some analysis should be presented to prove that at the minute scale rainfall statistics are adequately reproduced. Please provide more information on the analysis: how many observed years are there? 43? How many years are simulated for a given realization and how many realizations are composing the simulated ensemble?

[16 5] Varying from ensemble to ensemble. To overcome this - I suggest simulating an ensemble of 50 realizations of 43 years each.

[16 10-12] Belongs to the methodology section above. [Figure 9] Consider plotting the results spatially, i.e. over a map of the catchment. Then one can see if, for example,

СЗ

the biases are increasing moving toward the catchment boundaries, are depended on elevation, etc.

[19 15] What is 'n' here? The number of rainfall events?

[20 5] Peleg and Morin, 2014 - I think Peleg and Morin (2012) is a more suitable reference here.

---

## Referee Comment (RC2) · Anonymous Referee #2 · 29 Jun 2018

**Synopsis**

This paper presents STORM, a stochastic rainfall model using probability density functions of storms characteristics to generate synthetic individual storms over a grid. The model is flexible and the user has the possibility to perform case-control experiments by shifting the observed PDFs or adding an underlying trend to storm characteristics. The model has wide applications in the context of climate change.

**Major comments**

Generating synthetic time series of rainfall on a grid over short time steps is a very ambitious work. This is not an easy task and authors made a huge effort to develop

STORM. I was impressed. Regarding the form, the paper is very well-written and the figures are of high quality. I recommend this paper for publication after a few clarifications but I am sure that authors will address my concerns easily enough.

My main criticism is the lack of information on how the PDFs (storm area and storm center location) are derived from observed data. Some information are disseminated throughout the text (sections 2, 3 and 6) but the reader would like to know all the details as soon as possible in one single paragraph. My first question as a potential user is: how am I supposed to derive these PDFs from my network of rain-gauges? Authors have made a number of assumptions to define storm center and storm area and this warrants more clarification and justification.

Figure 1 This is a very nice figure but the reader needs more information. Panel b) why did you use a GEV fit? Sounds like a Gamma distribution would be a better fit although I may be wrong. Same question regarding the other panels. How did you select the distribution that best fits the data? An objective approach would be to compare different distributions using AIC or some sort of model selection criterion.

Figure 1c I find the term center confusing. To me, the center refers to the centroid. Please clarify.

Figure 1e Do you have an explanation for the stabilization of rainfall intensity between $10^1$ and $10^2$ minutes?

Figure 1f Please add a legend (if possible). Is the green curve indicating the 90th percentile as seen in Figure 1e?

The first component I was expecting to see in this figure is the storm frequency. Isn't it required to initialize STORM? Also, what about the diurnal cycle of rainfall events? Intense storms (and rainfall in general) are strongly locked onto the diurnal cycle. I was surprised to see no mention of the diurnal cycle of rainfall in the text.

Page 7 Line 22 Yes, but this issue is also expected in the other bins, right? I guess this

also depends on the traveling speed of the storm and the temporal resolution of the observed data used to derive the PDF.

Page 8 Line 24 Rainfall does vary with elevation but this variation also depends on wind direction (windward vs leeward sides). How does STORM deal with that? Authors acknowledge that a more explicit method is needed (I guess authors will investigate this point in further studies) but a couple more sentences would be welcome. For example, would it be possible to derive a PDF of intensity and duration based on a metric reflecting collectively orography, mean wind direction and mean wind speed? This is very relevant, especially when working at the watershed scale. Taking into account leeward vs windward effects would greatly improve STORM.

**Minor comments**

Page 1 Line 20 "We explain the how" should read "We explain how..."?

Page 2 Line 5 I am not sure that "spatial resolutions" is the most appropriate wording here. Rain-gauges are point measurements and are generally thought to provide information at the finest possible scale. The problem is rather their spatial representativeness, their limited coverage and low density. Also, another issue with most rainfall datasets is their coarse (typically daily) resolutions of measurements.

Page 2 Line 4-15 Could be relevant to mention radar datasets in this paragraph that may overcome some of these limitations.

Page 2 Line 8 Another general criticism is that most GCM are unable to simulate the diurnal cycle of rainfall extremes correctly but also basic features such as rainfall intermittency (Trenberth et al. 2017).

Trenberth, K. E., Zhang, Y., & Gehne, M. (2017). Intermittency in Precipitation: Duration, Frequency, Intensity, and Amounts Using Hourly Data. Journal of Hydrometeorology, 18(5), 1393–1412. https://doi.org/10.1175/JHM-D-16-0263.1

Page 2 Line 14-15 I don't understand why this information is challenging to summarize

over longer periods? Please clarify.

Page 3 Line 11 Why decadal? It would be useful to provide some information regarding the computational demands of STORM for a specific case study (e.g. the small watershed in SE Arizona).

Page 3 Line 19 What is exactly the storm center location? Is this the centroid of the storm or the rain gauge with the highest intensity?

Page 3 Line 20 How do you define storm duration? Is this the consecutive numbers of wet time steps or do you allow for a few dry time steps within the wet spell?

Page 3 Line 5-6 Can this be done using monthly PDFs? This would enable STORM to simulate rainfall more realistically for example in regions where the intensity is strongly modulated on sub-seasonal timescales (e.g., monsoonal regions). Some papers have shown that rainfall intensity is the highest during the onset of the monsoon when the soils are dry and when there is enough moisture to trigger deep moist convection.

Page 4 Line 32-33 The temporal resolution of the simulated rainfall depends on the temporal resolution of the historical data used to initialize STORM, right?

Page 5 Line 6 Unclear. Why the median? Shouldn't it be the sum?

Page 5 Line 7 "...PDFS for THIS THIS paper..."

Page 8 Line 6 is this the maximum seasonal/annual $P_I$?

Page 8 Line 19 Why did you constrain your attention to the first/last 3 curves? What about using a linear relation between the curve number and the probability of selection (e.g. curve 5 would have a lower probability of selection than curve 8)?

Figure 3 Very nice figure. I would suggest replacing the curve number by the percentile. This would be more informative for the reader. Also, what is the range of the OG1-lowest elevation group?

Figure 5 Please indicate on this figure the record length of observations (blue dots).

Page 11 Line 21 and Page 12 Line 4 "...a new PDF of for...". Is that right?

Page 12 First paragraph Can you clarify whether or not it is possible to add a trend in storminess without altering $P_{total}$. The user can be interested in simulating climate conditions where storms are getting more intense but less frequent without modifying $P_{total}$.

Figure 9 Authors acknowledge that STORM tends to overpredict storm total as well as annual P. Indeed, the amplitude of simulated values in much larger than that of observations. Can authors provide an explanation? The Monte-Carlo resampling procedure and the use of a large number of N-year simulations should produce distributions similar to the observed ones. This needs clarification.

Figure 10 Is this showing the grid point co-located with the Walnut Gulch watershed? It would be neat to add 2 subplots showing the decline in $P_I$ and the increase in $P_{total}$ seen in high-resolution observed data.

Page 18 Line 4-13 Another interesting application of STORM would be to characterize the network density needed to detect changes in short-duration rainfall extremes that are generally very localized in space (and thus often missed by surface stations). Given that daily (and longer) duration extremes are generally associated with larger systems, their detection from surface rain-gauges should be easier. If we progressively increase rainfall intensity in STORM, it could be interesting to sample a few fixed grid cells as hypothetical surface stations and see whether trends (or other statistics) emerge in short (minutes to hours) or long-duration (daily to multi-days) rainfall extremes first (see Kendon et al. 2018 and Barbero et al. 2017). This may provide insights on the network density needed to detect changes in rainfall characteristics.

Kendon, E. J., Blenkinsop, S., & Fowler, H. J. (2018). When will we detect changes in short-duration precipitation extremes? Journal of Climate.

https://doi.org/10.1175/JCLI-D-17-0435.1

Barbero, R., Fowler, H. J., Lenderink, G., & Blenkinsop, S. (2017). Is the intensification of precipitation extremes with global warming better detected at hourly than daily resolutions? Geophysical Research Letters, 44(2). https://doi.org/10.1002/2016GL071917

Page 20 Line 3-6 The density of rain-gauges is expected to affect the PDF of storm areas but also that of storm gradient.

Please also note the supplement to this comment:
https://www.geosci-model-dev-discuss.net/gmd-2018-86/gmd-2018-86-RC2-supplement.pdf

---

## Author Response (AR1)

**Responses to Reviewers Comments**

We thank Nadav Peleg and one anonymous reviewer for their careful review of our manuscript. We respond to their suggestions inline below. In addition, we have responded to the requests of the Executive Editor to provide the version number of our code and working links to the actual code and sample data.

Comments of Reviewer 1

Flow chart: We have provided a list of relevant distributions (now in Fig 2) and a description of both model initialization and model operation. Furthermore, the code itself is very well documented in terms of explaining precisely what the model does at teach stage. We have also added a new flow chart (new Fig 1).

Table of input parameters/distributions: These items are all well documented within our code, so we feel it is redundant to include them in the paper as well. We apologize if the reviewer did not have access to the code while creating his review.

New paragraph/other models: We have added reference to several other rainfall models and highlight the advances that STORM offers.

Re-analysis data: We have added clarification to our statement about re-analysis data products - This is especially true in regions where orography and other complicating land surface dynamics affect rainfall fields'. We feel this will address the reviewer's concerns about our previous comment on re-analysis data.

Two seasons: We take a seasonal approach to this modelling, since seasonal totals of rainfall are typically important considerations for long-term watershed planning. However, this is also required in our current approach in order to allow stochastic variations in both seasonal (or annual) totals, as well as the storm characteristics themselves. There is nothing stopping someone from further subdividing our model into 12 seasons, enabling monthly analysis of rainfall. The only inherent (and explicit) cross correlation in our model structure is the relationship between rainfall intensity and duration. We have not investigated the cross correlations structure of other model inputs.

Storm centers: No, there is only one storm center at a time in the current version of the model.

PET: PET is simulated a separate module within STORM. The reviewer is correct that it is not dependent on the rainstorm simulation. However, day and night value of PET are simulated separately, and these vary on a monthly basis (based on monthly data distributions).

500-m spacings: We removed reference to this in the general model description.

Link: We apologize that the link was not working. It is working in the updated version.

v.2017b: changed

Note: We moved the text to a more appropriate location.

Move text to methods: Done

Rainfall extremes: Our goal here is not to demonstrate that we can reproduce the spatial configuration of patterns in intensity. In fact, since intensity is a model input, we are not concerned about whether we have reproduced intensity as an output—it is a given. In fact, this is illustrated for the entire collection of gauging locations in Fig. 5. This figure shows that the model does indeed capture extreme rainfall events. We have added the length of each simulation (30 years) to Fig 9 for clarification.

Ensembles/simulations: We simulated 30 ensembles each of 30 years to evaluate STORM against observed rainfall data at WGEW. We have clarified this point in Fig 9.

Ensemble to ensemble: The reviewer has apparently misunderstood here. The simulation length and number of ensembles remains the same in all figures (30 ensembles each of 30 years). We were merely pointing out why values plot higher or lower than the 1:1 line (stochasticity).

Belongs to methodology: We disagree on this one. The data shown here are specifically from WGEW and we are demonstrating the model's skill in reproducing its rainfall characteristics.

Rainfall events: Yes, that is the 'n' referred to here. We've added clarification.

Reference: changed.

Comments of Reviewer 2

How were pdfs derived: Very good point. We have added clarification, including a reference to the automated distribution fitting tool recommended by the reviewer - For this paper, PDFs were fit manually using Matlab's Distribution Fitting Tool (distfittool), but we recommend that this be automated using a code that optimizes the fit based on maximum likelihood estimators: https://www.mathworks.com/matlabcentral/fileexchange/40167-fitmethis. Regarding which distributions are required, this information is contained in the comments of the code itself. We apologize if the link to the code was not working at the time of the review.

GEV v Other: We have added this info to the text – 'The particular distributions shown in Fig. 1 were generated by manual investigation based on best fit, but we recommend the automated approach.''

Fig 1c: We prefer 'storm center', since centroid refers to the mass of rainfall, rather than the areal center of the storm.

Stabilization: Unfortunately not. We have not investigated the processes of rainfall that lead to the particular functional form within the intensity-duration interdependence. However, we are currently working on these interdependencies using cupolas.

Fig 1f: It is not clear what the reviewer is asking for here, since there is already a legend on this figure panel. Yes, the darker of the two green curves is the 90th percentile of the relationship between intensity and duration.

Storm frequency/diurnal cycle: There is no explicit storm frequency included in the model. Instead its effect is captured by the number of storms simulated within a season, as well as by the simulated interarrival times. There is no diurnal cycle included here (apart from PET). Storms generated in our model can occur at any time with equal probability. We realize that many application might be concerned with the time of day in which the rainstorm occurred and there may be a selectivity to this, but we have not included this capability in STORM.

P7/L22: Yes, it is possible that this effect is also present in the other storm area bins. However, we believe (concurring with Syed et al) that the largest bin is the most affected and it is essentially spurious. Note: we have complete relied on the data from the Syed et al paper for this aspect of our application to Walnut Gulch.

P8/L24: Yes, wind speed and direction are likely to be relevant components of orography and we have added a statement of this in the text. We are currently working to improve characterization of orography in STORM using cupolas to capture the dependency between intensity and duration. We can also explore this for wind.

The how: changed

Spatial resolutions: changed to spatial representativeness

Radar: added mention

P2/L8: ref added

P3/L11: this was added to the sentence for clarification-- STORM performs this multi-layer parameter selection to create multiple sequences of spatially varying rainfall over a drainage basin and over a multi-decadal time series.

P3/L19: This was clarified above. The storm center is the actual middle of a circular storm area. It also has the highest intensity under our simulation method.

P3/L20: A rainstorm is characterized by a duration of non-dry days. Each one is punctuated by a randomly selected interstorm period.

P4/L5-6: Yes, this could be configured to run monthly (as 12 seasons). It would require some recoding, but it is certainly possible.

P4/L32-3: Yes, but one could certainly aggregate the native resolution of the rainfall data to a coarser resolution.

P5/L5: We think the reviewer has misunderstood here. The PTotal threshold is selected randomly and then the PTotals at every gauge are summed through the season until the median of all PTotals (at every gauge) exceeds the threshold value. We have clarified this in the manuscript.

P5/L7: Corrected

P8/L6: If the data exist, this can be done seasonally. We have clarified in the text.

P8/L19: We acknowledge that any rule-based approach like this is ripe for criticism. This is a short-term solution. We are currently working to improve the entire method for interdependency between intensity and duration.

Fig3: We looked into changing curve numbers to percentiles. However, we feel that this will create greater confusion, since we already have percentages of change to the probabilities listed on the figure. Instead, we have clarified in the caption that the curve numbers correspond to the percentiles as shown in (current) Fig 2E.

Fig 5: We have clarified this in the figure caption.

Of for: corrected

Trends in storminess: Yes, it is possible and we have added a sentence to clarify this point.

Fig 9: Yes, we acknowledge the slight overprediction in our simulated storm totals and annual precipitation totals. The figure only includes 30 ensembles of 30 years, so it is probable that with more simulations the centroid of values would shift back closer to the 1:1 line.

Fig 10: We have clarified this point in the caption. We note that the observations of declining intensity and increasing wetness at WGEW was already published elsewhere, so we have added a reference to that work.

P18/L4-13: This is a very interesting point, but beyond the scope of this paper. We should discuss collaboration on this using STORM with datasets of different temporal resolution. We have added this sentence to the manuscript: Another key area of future work would be to investigate how temporal resolution of rainfall data affects the signal of observed trends in rainfall (e.g., (Barbero et al., 2017)) and how these might yield different watershed responses.

Intensity gradient: Agreed. We have added this sentence - Finally, the density of the gauging network could have important influence on the storm intensity gradient with distance from the storm center, so the parameters of this relationship may be less certain for less dense gauge spacings.

---

## Author Response (AR2)

**Responses to Reviewer Comments**

[2 24] "even after downscaling". I suggest to delete this. After downscaling (which often also include a component of bias correction) climate at regional or local scales should be well represented.

DELETED.

[3 27] "1 or 2". Replace with "one or more", as you mention the model can be used on e.g. monthly basis.

REPLACED AS SUGGESTED.

[4 13] I don't think the link is necessary.

REMOVED.

[Figure caption 1] "source code documentation". Please add a reference to the document.

ADDED LINK.

[8 17-19] Repetitive. Can be deleted.

DELETED.

[new Figure 8] I suggest keeping this figure (ET simulation for 25 years) as Supplementary Information. I do not see the add value of it been with the main text.

WE DISAGREE WITH THE REVIEWER ON THIS POINT. WE BELIEVE IT IS IMPORTANT TO DEMONSTRATE WE CAN CHANGE THE INPUT ET WITHIN THE STORM CODE. PRESENTING THE FIGURE IN THE PAPER CLARIFIES WHAT EFFECT IT HAS ON THE SIMULATIONS.

[26 7-8] No need for a DOI here.

REMOVED HERE AND ELSEWHERE

[References] The list of references needs to be updated according to the references added/changed in the main text.

UPDATED.

[revised manuscript text omitted]